

# Transformer model-based multi-scale fine-grained identification and classification of regional traffic states

Jun Zhang and Guangtong Hu

School of Management and Engineering, Capital University of Economics and Business, Beijing, China

## ABSTRACT

To address the limitations in precision of conventional traffic state estimation methods, this article introduces a novel approach based on the Transformer model for traffic state identification and classification. Traditional methods commonly categorize traffic states into four or six classes; however, they often fail to accurately capture the nuanced transitions in traffic states before and after the implementation of traffic congestion reduction strategies. Many traffic congestion reduction strategies can alleviate congestion, but they often fail to effectively transition the traffic state from a congested condition to a free-flowing one. To address this issue, we propose a classification framework that divides traffic states into sixteen distinct categories. We design a Transformer model to extract features from traffic data. The k-means algorithm is then applied to these features to group similar traffic states. The resulting clusters are ranked by congestion level using non-dominated sorting, thereby dividing the data into 16 levels, from Level 1 (free-flowing) to Level 16 (congested). Extensive experiments are conducted using a large-scale simulated traffic dataset. The results demonstrate significant advancements in traffic state estimation achieved by our Transformer-based approach. Compared to baseline methods, our model exhibits marked improvements in both clustering quality and generalization capabilities.

## INTRODUCTION

The estimation of traffic states plays a pivotal role in traffic management and planning, significantly enhancing the efficacy of decision-making and resource allocation in Intelligent Transportation Systems (ITS) (*Olariu & El-Tawab, 2023*). Accurate and timely traffic state information is essential for optimizing traffic flow, reducing congestion, and improving overall road safety (*Yuan et al., 2014*). Predominant research endeavors in traffic states estimation have conventionally focused on utilizing traditional methodologies for quadripartite or hexapartite categorization of traffic states on individual roadways (*Ua-Areemitr, Sumalee & Lam, 2019*), falling short in addressing the nuanced requirements for regional traffic state identification and congestion reduction evaluation. Broad categorization can make it difficult to accurately assess the impact of traffic

Corresponding author
Guangtong Hu,
charles3000@cueb.edu.cn

congestion reduction strategies (*Cheng et al., 2020*). This article utilizes the Transformer model to extract features from traffic data. We then employ k-means clustering to partition the data into 16 clusters, and applying non-dominated sorting to hierarchically order the results of the clustering, forming congestion levels. This approach enables a more precise estimation of traffic states, addressing the demand for accuracy in traffic state estimation.

Accurate categorization of varying traffic states enables intelligent decision-making and efficient resource allocation (*Seo et al., 2017*). Traditional approaches to traffic state estimation often rely on sensor data and rule-based planning but are limited by data noise and the complexity of rule formulation (*Zhao et al., 2022*). Therefore, there is a demand for advanced methodologies that can effectively handle the complexities inherent in traffic data and provide more precise and reliable traffic information (*Min et al., 2023*). One of the most influential deep learning architectures in recent times is the Transformer model, which has revolutionized sequence modeling (*Lin, Wang & Lin, 2024*). We propose a framework that combines Transformer-based feature extraction with k-means clustering (*Zhao et al., 2023a*) and non-dominated sorting (*Bouarourou & Boulaalam, 2021*) to classify traffic states into sixteen distinct categories. Our methodology aims to leverage the inherent strengths of the Transformer model in adeptly capturing temporal and spatial dependencies within traffic data. The k-means clustering algorithm effectively groups similar traffic states into distinct clusters, while non-dominated sorting provides a structured hierarchy of these clusters based on congestion levels, enabling a more nuanced and comprehensive understanding of traffic conditions. By integrating the Transformer model with k-means clustering and non-dominated sorting, we surmount the limitations posed by conventional techniques.

In this article, we introduce a Transformer model specifically designed for capturing traffic state features, highlighting the use of multi-head self-attention and positional encoding. To further enhance the classification and ranking of traffic states, we integrate k-means clustering and non-dominated sorting into our framework. The k-means algorithm effectively groups similar traffic states into distinct clusters, while non-dominated sorting provides a structured hierarchy of these clusters based on congestion levels. To evaluate the effectiveness of our proposed approach, we conduct extensive experiments using large-scale real-world road network based simulated traffic datasets. We provide comparative assessments against traditional traffic state estimation methods. Additionally, we offer detailed analyses and insights into the interpretability and performance of the integrated framework comprising the Transformer model, k-means clustering, and non-dominated sorting.

Our contributions encompass the introduction of a novel traffic state estimation method that combines the Transformer model, k-means clustering, and non-dominated sorting. This method categorizes traffic states into sixteen distinct classes based on eight traffic indicators. Innovatively, we employ non-dominated sorting to hierarchically order the results of cluster analysis according to the severity of traffic congestion (*Deb et al., 2002*). This approach is thoroughly evaluated using large-scale simulated traffic datasets, showcasing its comprehensive effectiveness in traffic state estimation. Through this research, we aim to introduce a fresh, efficacious approach to traffic state estimation,

providing more precise data support for traffic management authorities. We are confident that our Transformer-based traffic state estimation method will furnish more accurate and optimize traffic policies through a comprehensive analysis of multiple traffic indicators.

The remainder of this article is organized as follows: "Literature Review" reviews relevant research in the field of traffic state estimation. "Materials & Methods" presents the experimental setup, problem statement, and modeling framework. "Results and Discussion" discusses the experiment results, providing insights into the performance of our models. "Conclusions" concludes the article, summarizing the key contributions, discussing the limitations, and outlining directions for future work.

## LITERATURE REVIEW

Traffic states estimation is a critical discipline in transportation engineering and urban planning, focusing on the retrospective examination of traffic states using historical data. It employs a variety of analytical tools, including statistical analysis, data mining, and machine learning, to extract patterns and correlations from static traffic datasets. This approach is essential for identifying long-term traffic trends, evaluating the impact of traffic policies, and forecasting future traffic states, thereby informing the development of sustainable transportation strategies and the optimization of existing infrastructure.

A series of studies have employed diverse mathematical models and statistical techniques to enhance the accuracy and robustness of traffic state estimation. *Varga et al. (2023)* implemented an enhanced Kriging interpolation method coupled with a novel data-driven distance measure to achieve efficient real-time prediction of urban traffic flow and velocity. *Cheng et al. (2024)* explored the uncertainty in traffic state transitions using a stochastic fundamental diagram model, validating its explanatory power over traffic phenomena with empirical data. *Rostami-Shahrbabaki et al. (2020)* introduced a two-layer model tailored for urban traffic network state estimation, leveraging connected vehicle data to augment prediction accuracy. *Wang, Papageorgiou & Messmer (2007a)* detailed a real-time highway traffic state estimation algorithm based on a stochastic macroscopic traffic flow model and an extended Kalman filter, enabling concurrent online estimation of key model parameters and traffic flow variables. *Kuwahara, Takenouchi & Kawai (2021)* highlighted the advantages of reverse-running probe vehicles in traffic state estimation during incidents. *Wang et al. (2022)* investigated the joint estimation of highway traffic state and model parameters utilizing a blend of fixed sensors and connected vehicle data. *Thai & Bayen (2014)* concentrated on the discrete hyperbolic scalar partial differential equation challenge in highway traffic state estimation, proposing an estimation technique rooted in interacting multiple models. *Shan & Zhu (2015)* constructed a traffic state network using GPS taxi data, optimizing the layout of urban traffic monitoring sensors. *Emami, Sarvi & Bagloee (2021)* optimized traffic signal control using connected vehicle information, employing Kalman filtering and neural network algorithms for predictive updates of traffic conditions. *Wang, Papageorgiou & Messmer (2007b)* specialized in real-time nonlinear estimation of highway traffic state, offering a method capable of online parameter estimation and adaptation to external changes.

*Kawasaki, Hara & Kuwahara (2019)*, *Jin & Ma (2019)*, and *Babu, Sure & Bhuma (2020)* integrated statistical methods with machine learning to address traffic state estimation challenges. *Kawasaki, Hara & Kuwahara (2019)* developed a state-space model fusing probe vehicle data with traffic flow models, applying sequential Bayesian filtering and cell transmission models for efficient multi-path traffic state estimation in two-dimensional networks. *Jin & Ma (2019)* formulated a general traffic state estimation framework based on Bayesian filtering, utilizing Gaussian process regression and extended Kalman filtering algorithms to deliver high-quality real-time state estimations across various traffic conditions and data sources. *Babu, Sure & Bhuma (2020)* employed sparse Bayesian learning (SBL) and block sparse Bayesian learning (BSBL) to handle undersampled freeway network data, demonstrating significant advantages in real-time traffic state estimation.

With the proliferation of deep learning technology, a succession of studies have successfully applied these techniques to tackle traffic state recognition challenges. *Zhang et al. (2024)* proposed a physics-informed deep learning (PIDL) model combined with computational graphs, effectively addressing traffic state estimation under data sparsity, showcasing superiority in various sparse data scenarios. *Xu et al. (2020)* presented a deep learning framework based on graph embeddings and generative adversarial networks, utilizing adjacent segment information for real-time road traffic state estimation, demonstrating its effectiveness in enhancing estimation accuracy. *Tu et al. (2021)* put forth a traffic flow state classification framework that leverages smartphone sensor data and an optimized deep belief network model for high-precision traffic state categorization. *Zhao & Yu (2023)* innovated the observer-informed deep learning (OIDL) method, combining partial differential equation observers with deep learning for precise spatiotemporal traffic state estimation, with experimental results affirming error reductions up to 30%. *Petrović et al. (2023)* introduced a hybrid soft computing model that blends two Gaussian conditional random field models, outperforming a variety of unstructured and structured models in large-scale traffic networks.

While existing research has advanced the development of ITS, three critical issues remain to resolve. First, the common practice of dividing traffic states into four or six categories often fails to adequately capture the nuanced effects of traffic congestion reduction strategies. Many interventions do not visibly transform traffic from congested to free-flowing, yet their impact on congestion is significant, necessitating a more refined categorization of traffic states. Second, while metrics such as speed, waiting time, and queue length play crucial roles in traffic state estimation, relying on single traffic parameters alone falls short in providing a comprehensive and accurate description of traffic conditions. Third, the results of the clustering process have not been systematically ordered according to the severity of congestion. These limitations collectively highlight the need for further refinement and expansion in current research efforts.

## MATERIALS AND METHODS

In this section, we introduce the road selection model based on multi-objective decision-making methods and complex network theory, as well as the Transformer based model for traffic state estimation.

## Definition of traffic state classification levels

In traditional traffic state estimation, traffic states are generally categorized into broad classes such as free flow, saturated flow, congested, and extreme congested. However, this broad categorization often fails to accurately reflect the complexities of real-world situations, particularly when evaluating the effectiveness of traffic congestion reduction strategies. Numerous types of traffic state classification levels have been proposed, including the Level of Service (LoS) in the United States, the TomTom Traffic Index (TTI) in Europe, and the Traffic Congestion Index (TPI) in China. *Mystakidis & Tjortjis (2020)* develops a decision tree model for predicting traffic congestion and categorizes the traffic congestion states into three classes. *Izhar, Quadri & Rizvi (2020)* uses support vector machine (SVM) and multinomial naive Bayes (MNB) classifiers based on the average speed of vehicles and the number of vehicles traveling on the road to perform binary classification of traffic congestion. *Zambrano-Martinez et al. (2017)* categorizes traffic states into four classes based on travel time for segments of the street through cluster analysis. *Toshniwal et al. (2020)* categorizes traffic states into six classes by performing spatiotemporal analysis on urban traffic data through clustering algorithms. This article proposes a multi-scale, fine-grained regional traffic state recognition and classification method based on the Transformer model. By subdividing traffic states into sixteen levels and integrating both macroscopic and microscopic indicators, this approach can more precisely capture the trends in traffic states.

The advancement in traffic microsimulation technology has rendered the acquisition of multiple traffic indicators for roads and vehicles within a given area both simple and cost-effective. Traffic indicators that are once challenging to measure, such as carbon monoxide (CO) emissions and noise, can now be readily obtained through traffic simulation. This article aims to categorize the traffic state $s_t$ based on a set of traffic indicators $T$, which includes average speed, average traffic flow, average delay, average number of stops, average queue length, average CO emissions, average travel time, and average noise. Given the varying importance of different road segments within a region, we employ complex network theory to select eight critical roads for data collection. After applying arithmetic averaging to the collected data, each row in the resulting dataset represents a single entry for the region, with each column corresponding to one of the eight mentioned indicators. Equation (1) presents the formula for identifying traffic states.

$$s_t = F(T) \tag{1}$$

where F denotes the model to be learned. In our research, we utilize the Transformer model to learn hierarchical traffic spatiotemporal features, and then apply k-means clustering and non-dominated sorting to achieve the identification of traffic states.

## Determination of traffic state classification metrics

A variety of metrics have been employed in prior research for traffic state estimation, encompassing macroscopic and microscopic indicators. Macroscopic indicators describe the operational characteristics of traffic, including flow, speed, density, occupancy, and queue length. Microscopic indicators elucidate the operational characteristics of vehicles

interrelated within the road network, such as travel time and delay. Generally, speed and flow are the most commonly utilized metrics (*Zhao et al., 2023a*, *2023b*).

The aim of this article is to classify the traffic state of real-world road networks. Previous researchers have used single indicators, such as speed or traffic flow, or combinations of two indicators to study this issue, but few studies have employed more than two indicators. Using a single or a few indicators to describe complex traffic conditions has limitations. Employing multiple indicators provides a more comprehensive reflection of actual traffic situations. In this article, eight indicators are utilized: speed, flow, delay, number of stops, queue length, CO emissions, travel time, and noise.

Traffic flow theory is a fundamental framework for understanding and modeling the movement of vehicles on roadways. It provides a basis for analyzing traffic conditions and predicting traffic behavior (*Zhai & Wu, 2021*). The three primary parameters in traffic flow theory are: speed, flow, density. Given that these parameters are mathematically interdependent, it is sufficient to select speed and flow as evaluation metrics. Speed and flow are traditional and intuitive ways to assess traffic smoothness, directly reflecting the efficiency of road utilization. Delay, average number of stops, average queue length, and travel time further refine the understanding of traffic flow, especially in the presence of traffic bottlenecks. Although CO emissions and noise levels are consequences of congestion, monitoring them can indirectly indicate the severity of traffic congestion.

## Selection of important road segments

When dealing with multiple road segments within a region, the identification of the overall traffic state faces the challenge of road redundancy. This article employs complex network theory to analyze the regional road network and combines multi-objective decision-making methods to assess and select the significance of road segments. The proposed research framework encompasses four criteria: degree centrality, betweenness centrality, closeness centrality, and eigenvector centrality (*Bamakan, Nurgaliev & Qu, 2019*). These criteria are commonly used in complex network theory to measure the importance of road segments. They are represented by Eqs. (2)–(5), respectively.

$$D(i) = \frac{\deg(i)}{n_1 - 1} \tag{2}$$

$$B(i) = \sum_{j \neq i \neq k} \frac{\sigma_{jk}(i)}{\sigma_{jk}} \tag{3}$$

$$C(i) = \frac{n_2 - 1}{\sum_{j \neq i} d(i,j)} \tag{4}$$

$$E(i) = \frac{1}{\lambda} \sum_{j=1}^{n_3} a_{ij} E_j \tag{5}$$

$D(i)$ is the degree centrality of node $i$, $deg(i)$ is the degree of node $i$, $n_1$ is the total number of nodes. $B(i)$ is the betweenness centrality of node $i$, $\sigma_{jk}$ is the number of shortest paths from node $j$ to node $k$, $\sigma_{jk}(i)$ is the number of those shortest paths that pass through

node $i$. $C(i)$ is the closeness centrality of node $i$, $d(i, j)$ is the length of the shortest path from node $i$ to node $j$, $n_2$ is the total number of nodes. $E(i)$ is the eigenvector centrality of node $i$, $\lambda$ is the largest eigenvalue of the adjacency matrix, $a_{ij}$ is an element of the adjacency matrix, $a_{ij} = 1$ if there is an edge between node $i$ and node $j$, and $a_{ij} = 0$, otherwise. $n_3$ is the total number of nodes, $E_j$ is the eigenvector centrality of node $j$.

After modeling various criteria related to the importance of road segments in the study area, it is necessary to aggregate them into a single index to rank the segments. In this study, the Entropy Weight-CRITIC (Criteria Importance Through Intercriteria Correlation)-TOPSIS (Technique for Order of Preference by Similarity to Ideal Solution) method is employed. Initially, the entropy weight method (*Zhu, Tian & Yan, 2020*) and CRITIC method (*Diakoulaki, Mavrotas & Papayannakis, 1995*) determine the influence weights of the four criteria on the importance of road segments, followed by the Technique for Order of Preference by Similarity to Ideal Solution TOPSIS method (*Yoon, 1987*) to rank the segments based on their significance.

### Entropy weight method

The entropy weight method is a widely used technique for calculating indicator weights, which measures the diversity and importance of indicators based on the concept of information entropy. Indicators with higher entropy values contribute more significantly to decision-making. Each indicator's data is normalized to ensure calculations occur under uniform dimensions. For a given indicator data matrix $D$, where rows represent samples and columns represent different indicators, the entropy value $e_j$ is calculated using Eq. (6).

$$e_j = -\frac{1}{\ln(n)} \sum_{i=1}^{n} p_{ij} \log_2 (p_{ij}) \tag{6}$$

where $n$ is the number of samples, and $p_{ij}$ is the proportion of the $i$-th sample for the $j$-th indicator, computed as in Eq. (7).

$$p_{ij} = \frac{x_{ij}}{\sum\limits_{i=1}^{n} x_{ij}}. \tag{7}$$

Based on the calculated entropy values, Eq. (8) shows the way to calculate weight $w_j$.

$$w_j = \frac{1 - e_j}{\sum\limits_{j=1}^{m}(1 - e_j)} \tag{8}$$

where $m$ is the total number of indicators, $j$ represents the indicator.

### CRITIC method

The CRITIC method is designed to calculate the correlation among indicators, measuring linear relationships based on a correlation coefficient matrix. Indicator data is standardized to transform it into a standard normal distribution with a mean of 0 and a standard deviation of 1. Pearson correlation coefficient is used to calculate the correlations between indicators. Based on the correlation coefficient matrix, a correlation matrix is further

calculated to measure the inter-indicator correlations, serving as a basis for weight adjustment.

(a) Calculation of weight adjustment factors

Suppose there are $n$ indicators, with a correlation matrix $C(n \times n)$ where $C(i,j)$ represents the correlation coefficient between the $i$-th and $j$-th indicators. Initialize the weight adjustment factor array $A(n \times 1)$ with all values set to 1. Then, for each indicator $i$, compute the sum of its correlations with other indicators by Eq. (9).

$$S(i) = \sum C(i, j). \tag{9}$$

Further, calculate the weight adjustment factor by Eq. (10):

$$A(i) = \frac{S(i)}{\sum S(j)} \tag{10}$$

(b) Final weight calculation

Assuming the initial weights are $W(n \times 1)$, initially calculated using the entropy weight method. Multiply the initial weights by the weight adjustment factors to obtain the final weights: $W_{final} = W * A$. The final weights reflect the relative importance and correlation of the indicators.

### TOPSIS method

The Technique for Order of Preference by Similarity to Ideal Solution (TOPSIS) method is a popular multi-criteria decision-making technique that ranks alternatives based on indicator weights and values. The determination of the Positive Ideal Solution (PIS) and the Negative Ideal Solution (NIS) fundamentally hinges upon the optimal and suboptimal performances of individual indicators. Based on their impact on traffic congestion, average speed and traffic flow are categorized as benefit indicators, signifying that higher values are desirable. Conversely, average delay, average number of stops, average queue length, CO emissions, travel time, and noise level are classified as cost indicators, indicating that lower values are preferable. The PIS is the ideal segment that maximizes values for each indicator, whereas the NIS minimizes them. For each segment, calculate its Euclidean distances to the PIS and NIS using Eqs. (11) and (12), respectively, measuring proximity to the ideal solutions.

$$D_i^+ = \sqrt{\sum \left(w_j \times \left(v_{ij} - V_j^+\right)\right)^2} \tag{11}$$

$$D_i^- = \sqrt{\sum \left(w_j \times \left(v_{ij} - V_j^-\right)\right)^2} \tag{12}$$

where $D_i^+$ indicates the distance between segment $i$ and the PIS, $D_i^-$ indicates the distance between segment $i$ and the NIS, $w_j$ is the weight of the $j$-th indicator, $v_{ij}$ is the standardized value of segment $i$ for the $j$-th indicator, $V_j^+$ is the standardized value of the PIS for the $j$-th indicator, and $V_j^-$ is the standardized value of the NIS for the $j$-th indicator.

Calculate the comprehensive score for each segment based on the distances to the PIS and NIS using Eq. (13):

$$S(i) = \frac{D_i^-}{D_i^- + D_i^+} \tag{13}$$

where $S(i)$ represents the comprehensive score of road segment $i$, with values closer to 1 indicating higher comprehensive scores.

### Ranking and selection of road segments

Rank the road segments according to their comprehensive scores and select those with higher rankings as important segments. Thresholds or a specific number of segments can be predefined based on specific needs.

By employing the TOPSIS method, we can rank and select road segments based on indicator weights and values, identifying critical segments. This facilitates focusing resources and improvements on the most significant segments under limited resources, enhancing the efficiency and robustness of traffic state estimation.

## Traffic state estimation model

Deep learning algorithms have demonstrated impressive results in the domain of traffic state estimation. This article introduces a traffic state estimation framework that integrates Transformers, k-means and non-dominated sorting. The self-attention mechanism of the Transformer enables parallel data processing, extracting spatiotemporal characteristics from traffic data to construct highly abstract and discriminative feature representations.

These features capture both local details and global patterns, making the Transformer an ideal choice for feature extraction in traffic state estimation. The incorporation of the K-Means algorithm serves to conduct clustering analysis on features encoded by the Transformer, summarizing complex traffic behaviors into sixteen typical state categories. The simplicity and effectiveness of k-means make it a practical tool for mapping high-dimensional features onto real-world traffic states, achieving dimensionality reduction while preserving sufficient information content. The application of non-dominated sorting serves as a critical step in hierarchically ordering the results of clustering. It transforms traffic state features into objective functions that reflect degrees of congestion, providing a structured way to rank and categorize the traffic states based on their severity. Figure 1 illustrates the data processing pipeline that transforms raw input data into congestion level. The process involves feature extraction using a Transformer, clustering with k-means, and final ranking through non-dominated sorting.

### Transformer feature extraction

A Transformer Encoder model is deployed to enhance feature processing. The network structure of the encoder is shown in Fig. 2. The multi-head self-attention mechanism is one of the core components of the Transformer model, allowing the model to simultaneously focus on different parts of the input sequence. The encoder layer receives data from the original input. First, the input data needs to be expanded in dimension to be compatible with the self-attention mechanism. Next, the multi-head self-attention mechanism is

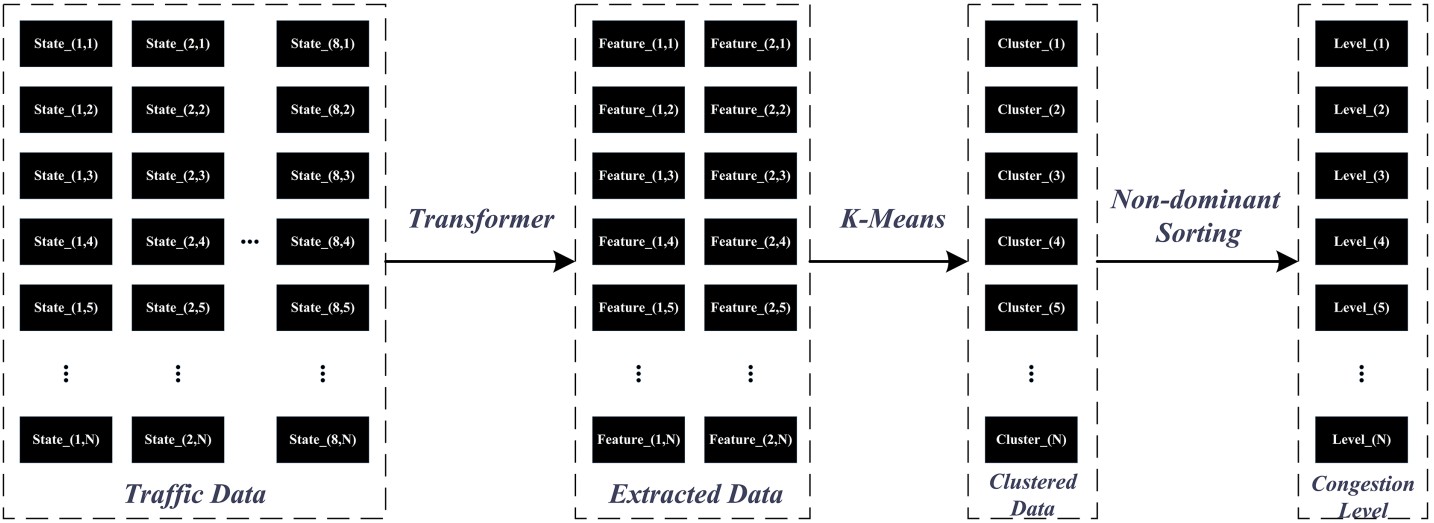

**Figure 1 Data processing framework for congestion level classification using Transformer, k-means, and non-dominated sorting.**

applied, with the number of attention heads set to 3, and the key and value dimensions for each head set to 8. The output of the self-attention mechanism is added to the input through a residual connection and processed with layer normalization to ensure stable information propagation. To prevent overfitting, a Dropout layer applies after the self-attention output.

The multi-head self-attention mechanism is a key innovation in the Transformer model. It works as follows: (1) The input data is first expanded in dimension to be compatible with the self-attention mechanism. This involves projecting the input data into multiple query (Q), key (K), and value (V) matrices; (2) For each attention head, the self-attention scores are calculated using the dot product of the query and key matrices, scaled by the square root of the key dimension, and then passed through a softmax function to obtain attention weights, as shown in Eq. (14); (3) The outputs from all attention heads are concatenated and linearly transformed to produce the final output of the self-attention mechanism.

$$\text{Attention}(Q, K, V) = \text{softmax}\left(\frac{QK^T}{\sqrt{d_k}}\right) V \tag{14}$$

where $d_k$ is the dimension of the key vectors.

Positional encoding is crucial for capturing the sequential information in time series data. We use fixed positional encoding to provide the model with absolute position information. First, a positional encoding matrix generates using sine and cosine functions. For each position *pos* and each dimension *i*, a positional encoding matrix is generated using sine and cosine functions by Eqs. (15) and (16), respectively. Then, this positional encoding matrix adds to the input embeddings, forming the final input representation. This way, the model not only processes the information contained in the input data but

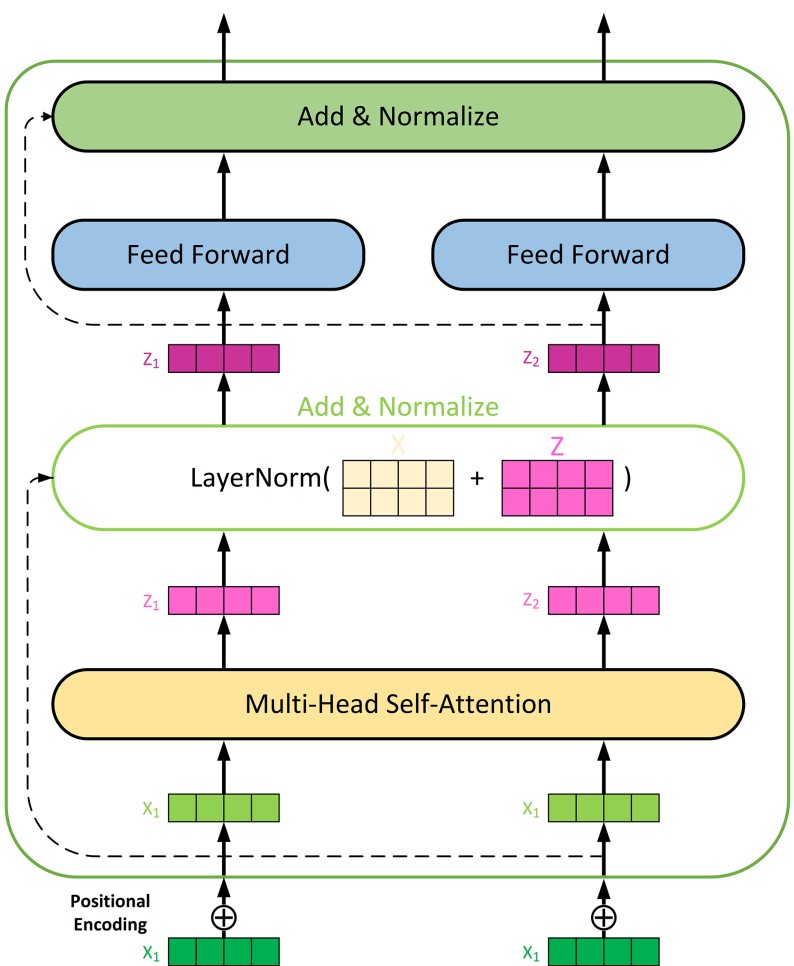

**Figure 2 Network architecture of the Transformer encoder for feature extracting.**

also captures the relative positional relationships within the sequence, thereby better understanding and handling sequential data.

$$PE_{(pos,2i)} = \sin\left(\frac{pos}{10000^{2i/dim}}\right) \tag{15}$$

$$PE_{(pos,2i+1)} = \cos\left(\frac{pos}{10000^{2i/dim}}\right) \tag{16}$$

where *dim* is the dimension of the input embeddings.

### K-means clustering

The k-means algorithm is applied to conduct clustering analysis on the features encoded by the Transformer. The algorithm flowchart of k-means is shown in Fig. 3. K-means algorithm performs clustering analysis on the features encoded by the Transformer. This process harmoniously integrates advanced feature representation with classical clustering techniques, ensuring both the accuracy and practicality of traffic state classification. K-means is an unsupervised learning algorithm used to partition a dataset into K clusters,

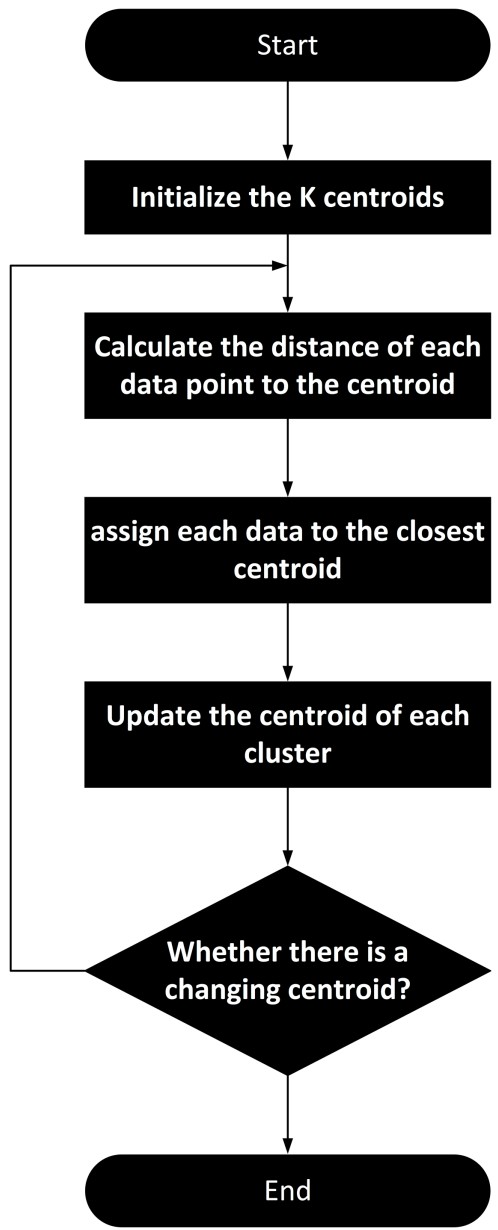

**Figure 3** **K-means clustering algorithm flowchart for clustering extracted feature data.**

where data points within each cluster are similar to each other, and data points in different clusters are dissimilar. The objective of the k-means algorithm is to minimize the Within-Cluster Sum of Squares (WCSS).

The steps involved in the k-means clustering process are as follows: (1) Randomly select K data points from the dataset as initial centroids; (2) Define the maximum number of iterations and the number of times centroids are initialized. In our implementation, the maximum number of iterations is set to 100,000, and the number of times centroids are initialized is set to 2,000; (3) For each data point, calculate its distance to each centroid and

assign it to the nearest centroid. Common distance metrics include Euclidean distance, Manhattan distance, and others. In our case, we use Euclidean distance, as shown in Eq. (17); (4) For each cluster, compute the new centroid as the mean of all data points assigned to that cluster; (5) If the centroids do not change significantly between iterations or if the maximum number of iterations is reached, the algorithm terminates. Otherwise, return to the Assignment Step.

$$d(x_i, c_j) = \sqrt{\sum_{k=1}^{d}(x_{ik} - c_{jk})^2} \tag{17}$$

where $x_i$ is a data point, $c_j$ is a centroid, and $d$ is the number of dimensions.

K-means receives two-dimensional features extracted from the Transformer layer and divides the data into 16 clusters. The choice of 16 clusters results from a balanced consideration of the need for fine-grained traffic state representation and training accuracy. This process harmoniously blends advanced feature representation with classical clustering techniques, ensuring accuracy and practicality in traffic state classification.

### Non-dominated sorting

The 16 clusters obtained through the k-means clustering algorithm do not directly reflect traffic congestion; they merely represent collections of data points with similar traffic states. Therefore, we use the non-dominated sorting algorithm to stratify all the data and correlate the clustering results with the levels of traffic congestion.

Non-dominated sorting is a widely used technique in multi-objective optimization. Its workflow includes initialization, calculation of dominance relations, layering, and output of the Pareto front list. Specifically, the input to non-dominated sorting is a set of solutions to a multi-objective optimization problem, where each solution consists of multiple objective function values (fitness values). The output is a set of non-dominated fronts, with each front containing a set of mutually non-dominated solutions.

In the initialization phase, non-dominated sorting receives a set of solutions as input. The next key step is the calculation of dominance relations. For two solutions A and B, if A is not worse than B in all objectives and is better than B in at least one objective, A dominates B. If no other solution dominates a particular solution, that solution is non-dominated. Based on this definition, non-dominated sorting first identifies all non-dominated solutions, forming the first front. Then, it removes the solutions in the first front and finds the non-dominated solutions among the remaining ones, forming the second front, and so on, until all solutions are assigned to different fronts. The final output is a list of fronts, with each front containing a set of mutually non-dominated solutions. The algorithm flowchart of non-dominated sorting is shown in Fig. 4.

In the hierarchical organization of traffic states, non-dominated sorting plays a crucial role. Traffic states can be described by multiple indicators, such as traffic flow and speed, which may conflict with each other. Through non-dominated sorting, a balance among multiple objectives is achieved, allowing the identification of different levels of traffic states. Each non-dominated front represents a set of traffic states with similar characteristics, and

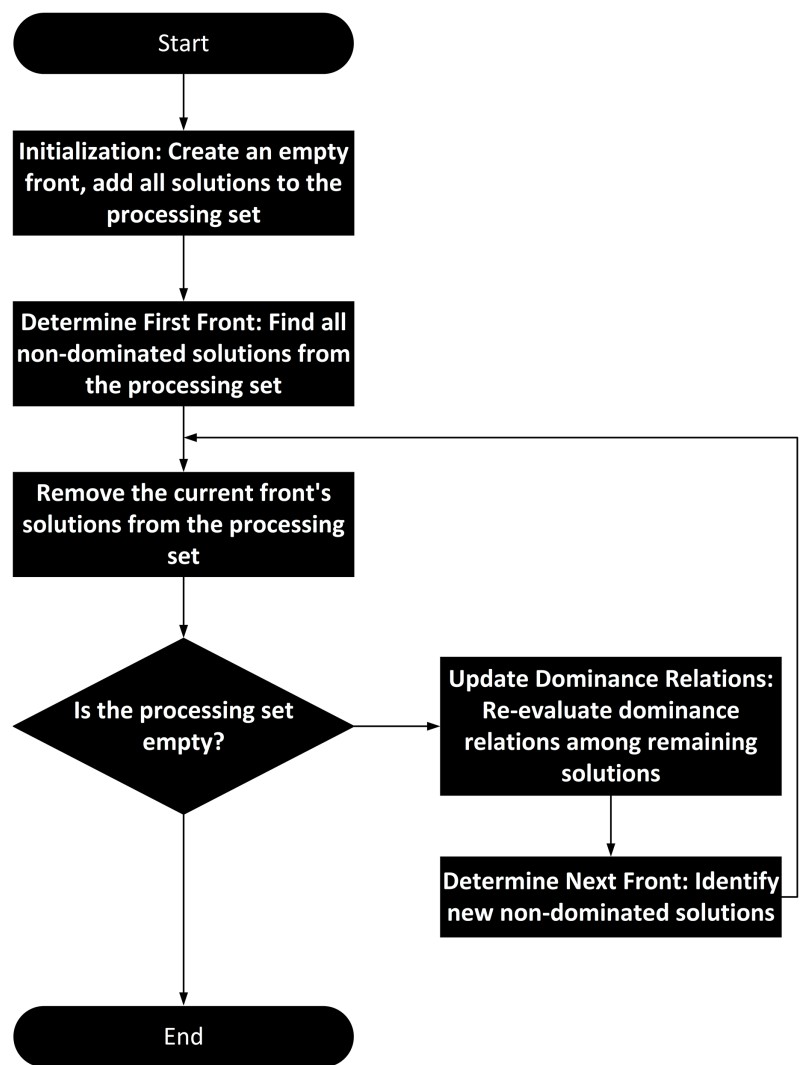

**Figure 4  Non-dominated sorting algorithm flowchart for mapping clustering results to congestion levels.**

the level of the front reflects the quality of the traffic state. Therefore, non-dominated sorting categorizes the original traffic state data into different levels, with each piece of raw data corresponding to a non-dominated rank.

K-means clustering divides the traffic states into 16 clusters, each representing a specific traffic pattern. The arithmetic mean of the non-dominated ranks of all data points within each cluster serves as the non-dominated rank for that cluster. Then, the clusters are sorted based on their non-dominated ranks to determine the congestion level for each cluster.

# RESULTS AND DISCUSSION

## Data preprocessing

We select Huitian area in Beijing, a region characterized by its diverse and complex traffic environment, as our study site. We conduct model training using an RTX 3060 (6 GB) on Windows 11 Operating System with Intel i7-12700H 2.30 GHz. In addition, the system

boasts a memory of 40 GB and utilizes high-speed Solid State Drives (SSDs) as its primary storage medium, with a total capacity of 1 Terabyte. SUMO (Simulation of Urban Mobility) is employed for road network construction and generation of simulation data to evaluate the performance of the traffic state estimation model (*Jiang, Ma & Koutsopoulos, 2022*). Using the Traci interface of SUMO, we are able to obtain real-time traffic data from the simulation environment.

In the data collection process, we have designed a series of logical judgment mechanisms to ensure the integrity and consistency of the data. For example, by setting reasonable thresholds and performing condition checks, we automatically detect and handle potential data anomalies, thereby preventing the occurrence of missing or abnormal values. Due to the implementation of these stringent logical judgments and data validation methods, our simulation data already possesses high quality at the collection stage. Consequently, in the subsequent data preprocessing steps, there is no need for additional handling of missing or abnormal values. This not only simplifies the data preprocessing workflow but also enhances the reliability of the data and the efficiency of model training.

The experimental road network is depicted in Fig. 5 (*Changping District People's Government of Beijing Municipality, 2022*). The roads for data collection are chosen based on the complex network theory discussed in "Materials & Methods". Eight critical roads, including the Jingzang Expressway, significantly impacting the local traffic situation are selected.

We use the duarouter tool provided by SUMO to generate vehicle routes on the real road network of the Huitian area. Duarouter randomly generates routes based on the set simulation time and the total number of vehicles, where the total number of vehicles is determined by multiplying the sum of vehicles traveling between all different origins and destination (OD) by a hyperparameter. To obtain a sufficiently diverse set of vehicle conditions, the simulation time is set to 2,400 h. Different OD pairs are selected from the road network, totaling 124, and the number of vehicles traveling between these OD pairs is randomly generated with a mean of 10 and a standard deviation of 5. It is noteworthy that, since the simulation occurs on a real road network, selecting such a large number of OD pairs aims to avoid the situation where only a few road segments have vehicles during the entire simulation. Additionally, to prevent the scenario where the number of vehicles is consistently too low or too high throughout the simulation, the parameters are repeatedly observed and adjusted. The final hyperparameter that determines the total number of vehicles in the network is set to 10,000.

Following data processing, we cut 600 data points for each traffic flow parameter. Figure 6 illustrates the temporal trends of averaged traffic characteristics, including average speed, average flow, average delay, average number of stops, average queue length, average CO emissions, average travel time, and average noise. The average speed is higher during off-peak hours and significantly lower during peak hours due to increased traffic volume. The average flow peaks during peak hours and is relatively low during off-peak hours. The average delay increases significantly during peak hours due to congestion, while it is minimal during off-peak hours. The average number of stops is more frequent during peak

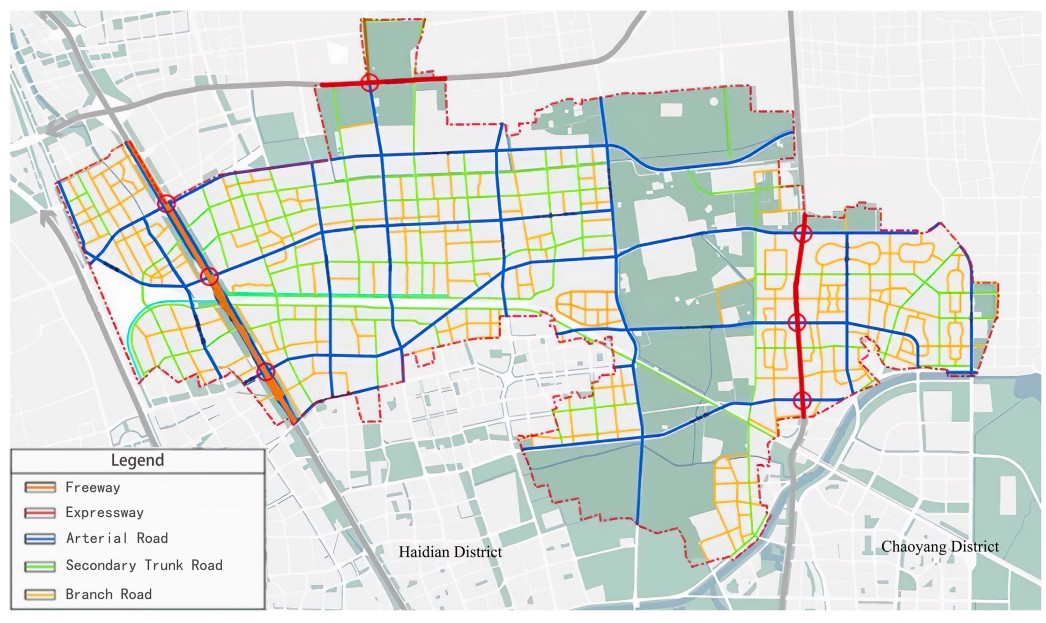

**Figure 5 Road network of the Huitian area with different road types marked.**

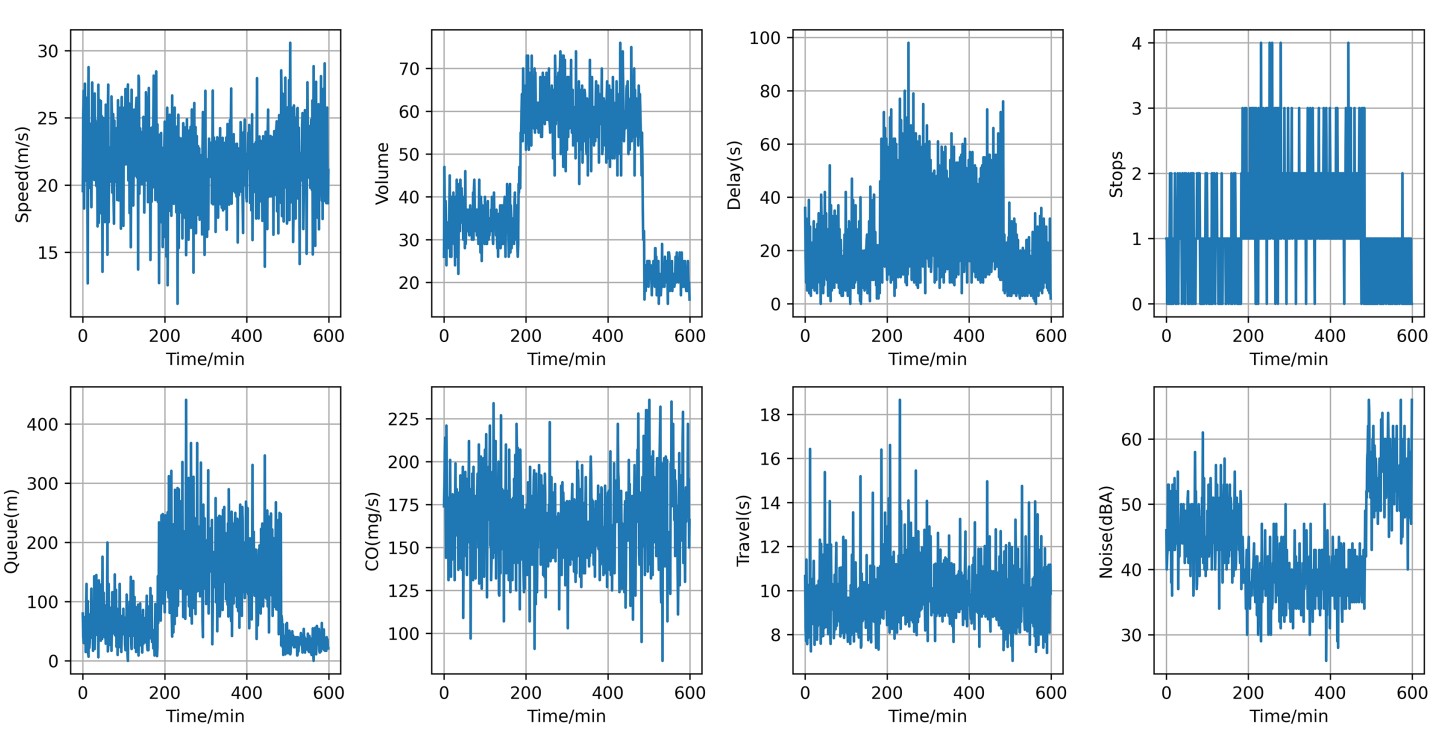

**Figure 6 The temporal trend of averaged traffic characteristics.**

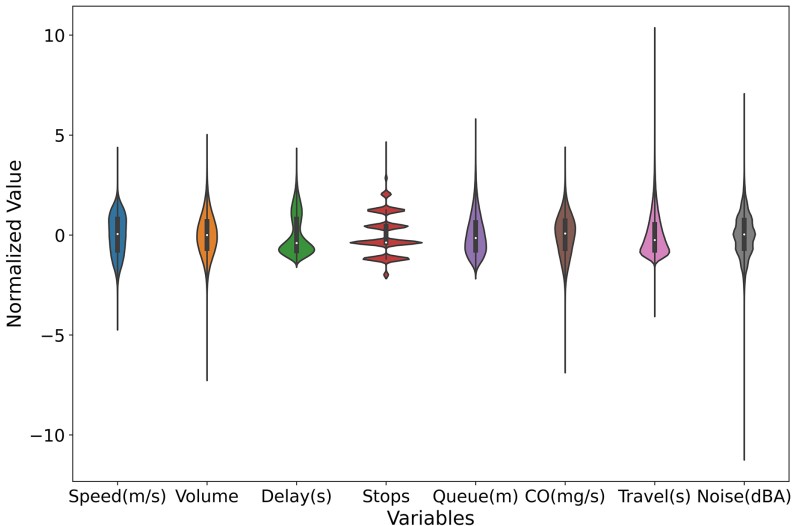

**Figure 7** Violin plots for multiple variables within the dataset.

hours due to traffic signals and congestion, and fewer during off-peak hours. The average queue length is longer during peak hours due to high traffic volumes and shorter during off-peak hours. The average travel time is longer during peak hours due to congestion and shorter during off-peak hours. CO emissions are higher during peak hours due to frequent acceleration, deceleration, and idling, and lower during off-peak hours. The average noise level is higher during peak hours due to frequent starts and stops, and lower during off-peak hours. These detailed time trends provide a comprehensive understanding of the traffic dynamics in the Huitian area, from efficient free-flowing conditions to severe congestion, and help identify the unique data features and environmental impacts associated with different traffic states.

Figure 7 presents violin plots for multiple variables within the dataset, each representing a distinct traffic metric, including average vehicle speed, average flow, average delay, average number of stops, average queue length, average travel time, average CO emissions, and average noise. These violin plots provide the distribution of each variable, offering a deeper understanding of the characteristics of the various traffic indicators. The distributions of average vehicle speed and flow are concentrated, indicating that vehicle speeds and traffic volumes remain fairly stable during the observation period. This suggests good road conditions and smooth traffic flow, with minimal frequent acceleration and deceleration. In contrast, the distributions of average delay, average number of stops, and average queue length are very compact, showing that vehicles experience consistent delay times, stop frequencies, and queue lengths. This consistency indicates a stable traffic condition without extreme delays. The distribution of average CO emissions is concentrated but shows some dispersion, suggesting that while most vehicles have similar CO emission levels, there is still some variability. This distribution may be influenced by factors such as vehicle type, driving behavior, and traffic conditions. The distributions of average travel time and average noise are highly dispersed, indicating significant variations

in the time required for vehicles to complete their journeys and in the noise levels experienced under different traffic conditions. Some vehicles complete their trips in a short time, while others take much longer. Similarly, noise levels vary widely, with some periods being quiet and others being quite noisy. This high dispersion reflects the instability of traffic conditions and the impact of various factors on travel time and noise. Overall, the violin plots reveal that the distributions of noise and travel time are highly dispersed, indicating significant variability in these metrics across different scenarios. The distributions of CO emissions, speed, and flow are concentrated but show some dispersion, indicating that while these metrics are stable, they still exhibit some variability. The distributions of average delay, average number of stops, and average queue length are very compact, indicating that these metrics remain relatively stable with little variation. These insights provide valuable references for traffic management and planning.

The dataset offers a holistic view of traffic and environmental indicators, encapsulating eight pivotal aspects: average vehicle speed, vehicle count, delay duration, number of stops, queue length, CO emissions, travel time, and noise level. Table 1 delineates the results of the statistics performed on the dataset. The average vehicle speed is approximately 21.42 m per second, with a standard deviation of 5.16, illustrating moderate velocity fluctuation. The robustness of the average speed estimate is underscored by a standard error of just 0.0136, indicating high precision. The vehicle count averages 40.41 vehicles, exhibiting substantial variability (standard deviation of 43.78), yet the reliability of the mean is maintained with a standard error of 0.1154. Delay duration (averaging 34.22 s) and travel time (averaging 13.40 s) may seem modest numerically, but their high standard deviations (108.05 and 222.14, respectively) and standard errors (0.2847 and 0.5854) reveal considerable inconsistency in these metrics, suggesting wide-ranging impacts on travel efficiency. The average number of stops (1.39) and queue length (175.13 m) display relative stability, albeit the high variability in queue length (standard deviation of 600.07) warrants attention, highlighting potential bottlenecks or congestion points that require management. CO emissions (averaging 165.54 milligrams per second) and noise levels (averaging 47.95 dBA) showcase high estimation accuracy and stability, with standard errors of 0.1054 and 0.0346, respectively, indicating reliable measurement consistency.

To represent the significance levels of the metrics used for classifying traffic states, we employ the t-test method. Table 2 delineates the results of the t-tests performed on the dataset. Specifically, a one-sample t-test is conducted at a 95% confidence level to assess whether the means of these metrics significantly deviate from zero. It is imperative to note that prior to conducting the t-test, the normal distribution and homogeneity of variances of these metrics should be verified, as these are foundational assumptions for the validity of t-tests.

The data analysis focuses on eight pivotal traffic and environmental metrics: average vehicle speed, flow, delay, number of stops, queue length, CO emissions, travel time, and noise level. The primary objective is to test the hypothesis that the means of these metrics do not significantly differ from zero (test value = 0). The outcomes reveal that the means of all metrics, along with their corresponding 95% confidence intervals, are substantially

**Table 1 One-sample statistics of the classification indicators.**

|  | Mean | Std. deviation | Std. error mean |
|---|---|---|---|
| Average speed (m/s) | 21.4209 | 5.1556 | 0.0136 |
| Average number of vehicles | 40.4073 | 43.7871 | 0.1154 |
| Average delay (s) | 34.2241 | 108.0516 | 0.2847 |
| Average number of stops | 1.3868 | 2.9497 | 0.0078 |
| Average queue length (m) | 175.1299 | 600.0682 | 1.5813 |
| Average CO emission (mg/s) | 165.5424 | 39.9848 | 0.1054 |
| Average travel time (s) | 13.4028 | 222.1414 | 0.5854 |
| Average noise level (dBA) | 47.9466 | 13.1478 | 0.0346 |

**Table 2 One-sample t-test of the classification indicators.**

|  | Test value = 0 | | | | |
|---|---|---|---|---|---|
|  | t | p | Mean difference | 95% Confidence | |
|  |  |  |  | Lower | Upper |
| Average speed (m/s) | −0.28 | 0.78 | 21.42 | 21.39 | 21.45 |
| Average number of vehicles | −0.45 | 0.65 | 40.41 | 40.18 | 40.63 |
| Average delay (s) | 1.04 | 0.30 | 34.22 | 33.64 | 34.78 |
| Average number of stops | 0.35 | 0.73 | 1.36 | 1.37 | 1.40 |
| Average queue length (m) | 0.21 | 0.84 | 175.13 | 172.03 | 178.23 |
| Average CO emission (mg/s) | −0.85 | 0.39 | 165.54 | 165.31 | 165.72 |
| Average travel time (s) | 0.99 | 0.32 | 13.40 | 12.26 | 14.55 |
| Average noise level (dBA) | 0.73 | 0.46 | 47.92 | 47.85 | 48.01 |

distant from zero, not straddling it, which strongly indicates that the observed means are significantly different from zero, not merely due to random variation.

In particular, the average vehicle speed (21.4209 m/s) yields a t-statistic of −0.2788 and a $p$-value of 0.7804, affirming that the mean vehicle speed is significantly greater than zero, with the confidence interval (21.3943 to 21.4475) entirely residing in the positive domain. Similarly, the means of flow (40.4073), delay (34.2241 s), number of stops (1.3868), queue length (175.1299 m), CO emissions (165.5424 mg/s), travel time (13.4028 s), and noise level (47.9466 dBA) are all markedly distinct from zero. Their respective t-statistics (−0.4491 to 0.9858) and $p$-values (0.3001 to 0.8371) corroborate the substantial divergence from the null hypothesis.

Notably, the 95% confidence intervals for all metrics fail to intersect with zero, reinforcing the statistical significance of the observed means differing from zero. This finding underscores the reliability of the observed mean values, indicating that they are not simply random occurrences but rather reflective of genuine traffic and environmental conditions.

## Assessment of traffic state estimation

In this study, the Transformer model is employed to extract features from traffic indicators. Subsequently, the k-means algorithm is utilized to cluster traffic states. Given the research objective to furnish a refined assessment of congestion alleviation effects for traffic congestion reduction studies, the number of clusters for traffic state is set to sixteen, signifying sixteen distinct levels of traffic states.

Figure 8 illustrates the comparative visualization of clustering outcomes between the principal component analysis (PCA)-based k-means clustering algorithm (as shown in Fig. 8A) and the k-means clustering augmented with Transformer algorithm (as shown in Fig. 8B), where both Transformer and PCA techniques reduce the original data dimensions to two dimensions for ease of graphical representation. Upon scrutinizing the depicted figures, it becomes evident that the k-means clustering augmented with Transformer algorithm demonstrates superior efficacy in segregating traffic data into sixteen distinct classes compared to the PCA-based k-means clustering algorithm. The PCA parameters are set as follows: number of components is 2, whitening is false, SVD solver is 'auto', tolerance is 0.0, and random state is none. The FCM parameters are set as follows: number of clusters is 16, initial centers initialized using k-means++ method, and fuzziness is 2.0.

We conduct a detailed comparative analysis of four different clustering methods: Transformer combined with k-means (TRANSFORMER+KMEANS), PCA coupled with k-means (PCA+KMEANS), fuzzy C-means clustering (FCM), and Transformer mechanism alone (TRANSFORMER). Using three primary quality metrics: silhouette coefficient, Davies-Bouldin index, and Calinski-Harabasz index. Table 3 delineates the performance comparison of four clustering algorithms across three quality metrics.

Firstly, the silhouette coefficient, a metric that quantifies how closely individual data points belong to their assigned cluster relative to other clusters. For a single sample $i$, the silhouette coefficient $s(i)$ is calculated as Eq. (18). Among them, $a(i)$ is the average distance between sample $i$ and all other points in the same cluster. $b(i)$ is the average distance between sample $i$ and all points in the nearest other cluster. The silhouette coefficient of the whole dataset is the average of the silhouette coefficients of all samples is calculated as Eq. (19).

$$s(i) = \frac{b(i) - a(i)}{\max(a(i), b(i))} \tag{18}$$

$$\text{Silhouette coefficient} = \frac{1}{n}\sum_{i=1}^{n} s(i) \tag{19}$$

TRANSFORMER+KMEANS as the frontrunner with a score of 0.6943074. This high value indicates that the clusters generated are internally cohesive and well-separated from each other, demonstrating an ideal clustering structure. The PCA+KMEANS method, with a silhouette coefficient of 0.508868418, also exhibits reasonable cluster quality but falls short of TRANSFORMER+KMEANS in terms of intra-cluster similarity and inter-cluster distance. FCM, scoring a meager 0.0609553, reveals significant issues with homogeneity

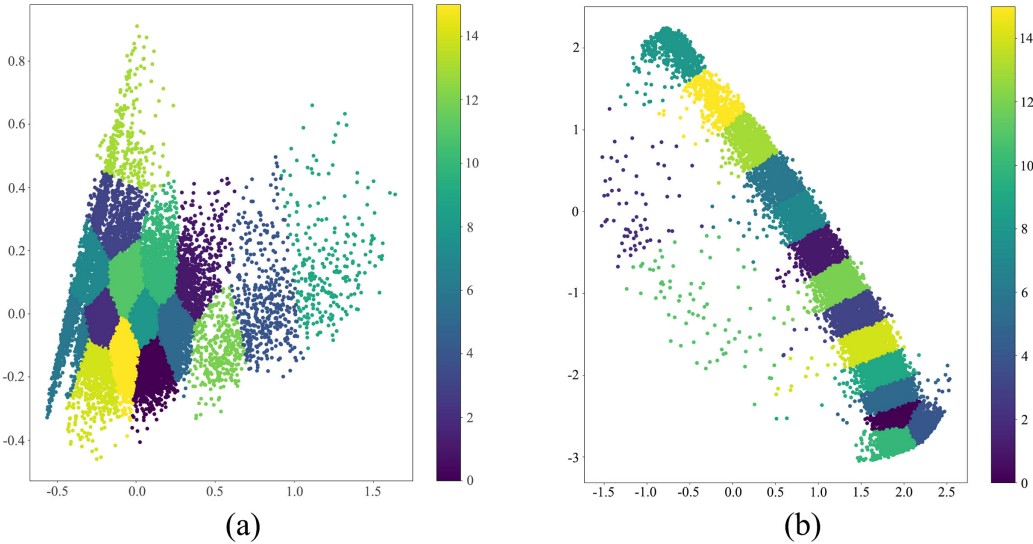

(a)  (b)

**Figure 8 Performance comparison analysis of transformer-enhanced _vs._ PCA-aided k-means in traffic states estimation.**

**Table 3 Performance comparison of four clustering algorithms across three quality metrics.**

|  | Silhouette coefficient | Davies-Bouldin | Calinski-Harabasz |
|---|---|---|---|
| TRANSFORMER+KMEANS | 0.6943 | 0.5429 | 719,083.31 |
| PCA+KMEANS | 0.5089 | 0.7635 | 23,479.98 |
| FCM | 0.0610 | 2.0931 | 543.21 |
| TRANSFORMER | 0.3901 | 5.0135 | 15,902.33 |

within clusters and differentiation between them, indicating subpar clustering effectiveness. TRANSFORMER, achieving a silhouette coefficient of 0.39005315, performs moderately, placing it between PCA+KMEANS and FCM, suggesting that the TRANSFORMER Mechanism alone, without k-means assistance, yields average results in clustering tasks. Secondly, the Davies-Bouldin index, which measures cluster separation by comparing the similarity of each cluster with all other clusters. Equation (20) defines a metric $d_{ij}$, for each cluster $i$ and cluster $j$, $s_i$ is the average distance of all points within cluster $i$ to the cluster center, $s_j$ is the average distance of all points within cluster $j$ to the cluster center, $d(c_i, c_j)$ is the Euclidean distance between the center of cluster $i$ and cluster $j$. For each cluster $i$, find the cluster $j$ that maximizes $d_{ij}$, and then calculate the average of all $d_{ij}$ values. Then, the Davies-Bouldin index is given by Eq. (21).

$$d_{ij} = \frac{s_i + s_j}{d(c_i, c_j)} \tag{20}$$

$$\text{Davies-Bouldin Index} = \frac{1}{k} \sum_{i=1}^{k} \max_{j \neq i} d_{ij} \tag{21}$$

TRANSFORMER+KMEANS with the low Davies-Bouldin index of 0.542877922 signals smaller average distances within clusters and larger distances between clusters, indicative of high-quality clustering. PCA+KMEANS, with a slightly higher Davies-Bouldin index of 0.76353081, still maintains a favorable position, suggesting good cluster differentiation. Conversely, FCM's elevated Davies-Bouldin index of 2.093102144 and Transformer's exceedingly high index of 5.013490372 strongly suggest poor cluster separation and high intra-cluster similarity, respectively, highlighting their inferior performance in this regard.

Lastly, the Calinski-Harabasz index, measuring the ratio of between-cluster dispersion to within-cluster dispersion. Equation (22) defines the Calinski-Harabasz index, which $Tr(B_k)$ is the trace of the between-cluster scatter matrix and $Tr(W_k)$ is the trace of the within-cluster scatter matrix. Among them, n is the total number of samples and k is the number of clusters. Equations (23) and (24) defines $Tr(B_k)$ and $Tr(W_k)$ respectively. Where $n_i$ is the number of samples in cluster $i$, $m_i$ is the center of cluster $i$, and $m$ is the center of all samples.

$$Calinski - Harabasz\ Index = \frac{Tr(B_k)}{Tr(W_k)} \times \frac{n-k}{k-1} \tag{22}$$

$$\mathrm{Tr}(B_k) = \sum_{i=1}^{k} n_i \|\mathbf{m}_i - \mathbf{m}\|^2 \tag{23}$$

$$\mathrm{Tr}(W_k) = \sum_{i=1}^{k} \sum_{x \in c_i} \|x - \mathbf{m}_i\|^2 \tag{24}$$

TRANSFORMER+KMEANS with the Calinski-Harabasz index of 719083.3145 denotes a substantial disparity between inter-cluster and intra-cluster variances, signifying exceptional clustering effectiveness. PCA+KMEANS, with a Calinski-Harabasz index of 23479.98151, lags behind but still showcases commendable clustering ability. FCM, scoring a mere 543.2060305, and TRANSFORMER, with a 15902.32698 index, fall significantly lower, implying less effective clustering due to smaller ratios of between-cluster to within-cluster dispersion.

In conclusion, TRANSFORMER+KMEANS emerges as the clear winner across all evaluation criteria, displaying outstanding clustering outcomes. PCA+KMEANS also exhibits decent performance in some metrics, while FCM and TRANSFORMER alone lag considerably behind, especially FCM, which consistently scores the lowest in all indices, underscoring its least favorable clustering efficacy. Hence, for achieving optimal clustering quality, TRANSFORMER+KMEANS should be the preferred choice among the evaluated methods.

## Evaluation of traffic decongestion impact

The assessment of traffic congestion reduction effects is pivotal for evaluating the efficacy of carpooling strategies. Utilizing common metrics in multi-objective optimization algorithms, dominance and crowding distance, we rank and distinguish congestion levels resulting from clustering analyses.

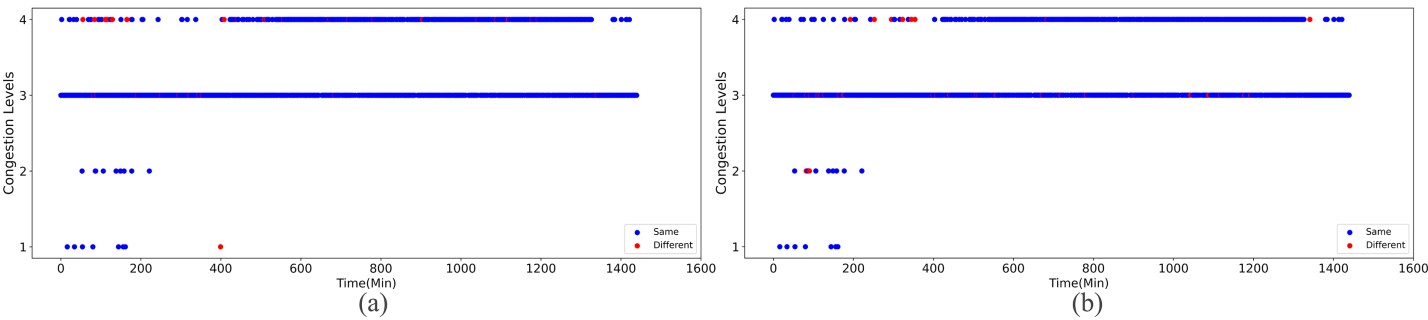

**Figure 9 The temporal trend of traffic state changes in the four-class model before and after the implementation of traffic mitigation measures.**

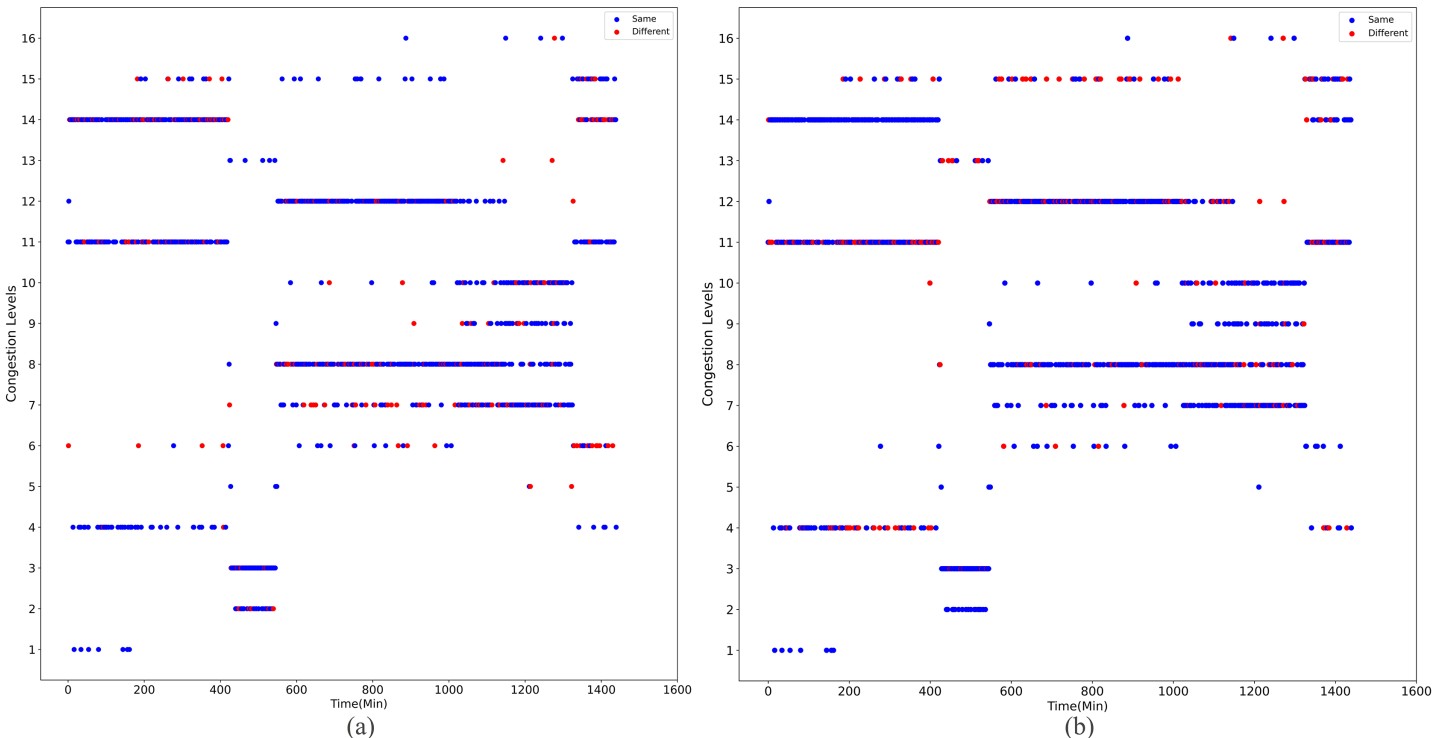

**Figure 10 The temporal trend of traffic state changes in the sixteen-class model before and after the implementation of traffic mitigation measures.**

Upon implementing decongestion strategies in the road network, a comparative illustration is presented in Figs. 9 and 10. They comprehensively depict the temporal evolution of traffic conditions predicted by two models, a four-class model and a sixteen-class model, both before and after intervention. The figure comprises two subplots for side-by-side contrast, where Figs. 9A and 9B chronicle the temporal trend of traffic state changes in the four-class model before and after the implementation of traffic mitigation

measures, respectively. Similarly, Figs. 10A and 10B display the corresponding dynamics for the sixteen-class model under the same conditions.

Post-strategy implementation, the discrepancies become starkly evident: The four-class model registers a mere 35 instances of traffic state transitions, indicating limitations in capturing nuanced variations in decongestion impacts. Conversely, the sixteen-class model identifies a staggering 273 shifts in traffic states, underscoring its superior potential for fine-grained discrimination and evaluation of traffic improvement outcomes. These shifts encompass not only transitions from severe to mild congestion but also subtle enhancements that might elude coarser classification systems.

Thus, the series of diagrams not only attests to the superiority of the sixteen-class model in delineating congestion with greater precision but also implies that, in tackling the intricate challenges of urban traffic flow management, adopting higher-resolution classification schemes is critical for accurately monitoring the effects of decongestion strategies and devising efficacious responses.

## CONCLUSIONS

This article introduces an innovative traffic state identification method based on the Transformer model, effectively overcoming the limitations of traditional approaches in terms of granularity. By implementing a novel sixteen-level categorization and integrating both macroscopic and microscopic indicators, it achieves a sophisticated differentiation of traffic conditions. The approach also leverages complex network theory to identify crucial road segments and applies the Entropy Weight-CRITIC-TOPSIS method to assess their significance, thereby enhancing the efficiency of data handling processes. Non-dominated sorting algorithms are employed to rank the clustering outcomes by k-means algorithm. Large-scale simulation data experiments validate the superiority of the model.

In practical applications, beyond issues pertaining to the evaluation of congestion alleviation measures, an excess of classifications might result in augmented operational intricacy, potentially impairing policymakers' capability to react promptly. While the model's training and validation rely on large-scale simulated traffic datasets that offer rich and controllable experimental environments, these may not fully capture the complexity and unpredictability of real-world traffic conditions. Consequently, future work will focus on validating the model with real-world data, integrating it with other deep learning architectures, and developing real-time traffic management systems. These efforts aim to enhance the model's robustness, accuracy, and practical applicability, ultimately contributing to more efficient and effective urban traffic management.

### Funding

This study is supported by grants from the National Social Science Foundation of China (grant number 20BGL001). The funders had no role in study design, data collection and analysis, decision to publish, or preparation of the manuscript.

## Grant Disclosures

The following grant information was disclosed by the authors:
National Social Science Foundation of China: 20BGL001.

## Competing Interests

The authors declare that they have no competing interests.

## Author Contributions

- Jun Zhang conceived and designed the experiments, authored or reviewed drafts of the article, and approved the final draft.
- Guangtong Hu conceived and designed the experiments, performed the experiments, analyzed the data, performed the computation work, prepared figures and/or tables, authored or reviewed drafts of the article, and approved the final draft.

## Data Availability

The raw measurements are available in the Supplemental Files.

## Supplemental Information

Supplemental information for this article can be found online at http://dx.doi.org/10.7717/peerj-cs.2625#supplemental-information.

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
