# Peer review of "Transformer model-based multi-scale fine-grained identification and classification of regional traffic states"

_PeerJ Computer Science, doi:10.7717/peerj-cs.2625_

## Round 0.1 · original submission · Major Revisions

Dear authors,

Thank you for submitting your article. Based on reviews' comments, your article has not yet been recommended for publication in its current form. However, we encourage you to address the concerns and criticisms of the reviewers and to resubmit your article once you have updated it accordingly. Before submitting the paper, following should also be addressed:

1. Equations should be used with correct equation number. Please do not use “as follows”, “given as”, etc. Explanation of the equations should also be checked. All variables should be written in italic as in the equations. Their definitions and boundaries should be defined. Necessary references should be provided.
2. Many of the equations are part of the related sentences. Attention is needed for correct sentence formation.
3. Pay special attention to the usage of abbreviations. Spell out the full term at its first mention, indicate its abbreviation in parenthesis and use the abbreviation from then on.
4. All of the values for the parameters of all algorithms selected for comparison should be provided.

Best wishes,

·

Basic reporting

- The Introduction adequately introduce the subject and make it clear, but only needs some references
- Literature well referenced & relevant.

Experimental design

- "Methods are not described with sufficient detail"
- There a discussion on data preprocessing and is it sufficient
- Sources adequately included.

Validity of the findings

- The experiments and evaluations are not performed satisfactorily, there is a need for travel time evalution versus time
-The Conclusion identify unresolved questions, limitations, and future directions

Additional comments

In this work, the authors proposed a novel approach based on the Transformer model for traffic state identification and classification to address the limitations in precision of conventional traffic state estimation methods. They introduced some examples of failures for other methods, including transitions in traffic conditions before and after the implementation of decongestion strategies, strategies do not effectively shift the state from congested to free-flowing ... For these raisons, the authors designed a Transformer-based model architecture to extract features from traffic data. Subsequently, K-Means clustering is applied to these features to group similar traffic states.

I find both considered topic and solution in this work more interesting. Also, the paper is well organized and structured ; however there are some drawbacks to take into consideration, including:

1) The main drawback of this paper is related to an absence of sufficient details about the proposed meteorology as the authors provided only some few details. The authors invited to add theoretical part, some explaining flow charts, illustrations and algorithms, simulation parameters to help readers understand the strong aspects of proposed solution.

2) In the introduction part:
The whole introduction is based only on three references, which is considered a bad practice to determine the context of this work. The authors are invited to add some reference in introduction part, especially recent ones to reflect the novelty of the problem and solution.

3) In the Problem Statement part: the authors realized this part without considering some references to help readers understand the problem, especially the novelty of the problem.

4) In the whole paper, there is a need for some references: the authors are invited to add some references to reflect the novelty of this paper as compared with existing works of the same context.

5) In evaluation part: there is a need for simulation parameters used for SUMO (Simulation of Urban Mobility) and equations used for performance metrics considered in this work.

6) In evaluation part: It's better make some plots vs time for Average Travel Time as this performance metric is so importance to reflect other related performance metrics like speed ...

·

Basic reporting

Good!

Experimental design

Good!

Validity of the findings

Good!

Additional comments

Firstly, I would like to congratulate the authors for the excellent work so far, below are some suggestions to qualify the research, namely:

- I suggest inserting one more topic, right after the Conclusion, addressing what is expected from this work in the future. This aspect is essential as it demonstrates the importance of the present study for the future;

Reviewer 3 ·

Basic reporting

This paper introduces a transformer-based traffic state prediction model. Although the study claims its contribution lies in using complex models and multiple variables, it lacks sufficient justification for these claims. Please refer to the comments below:
1. The introduction should include the importance and necessity of traffic state estimation. Review a broader range of studies comprehensively and make improvements accordingly.
2. Similarly, the literature review lacks coverage of the state-of-the-art in traffic flow. This section needs more depth and inclusion of recent advances in the field. Here are examples: A continuous traffic flow model considering predictive headway variation and preceding vehicle's taillight effect; A continuum model considering the uncertain velocity of preceding vehicles on gradient highways; Congestion boundary approach for phase transitions in traffic flow.
3. Additionally, studies related to traffic state prediction, such as those involving XAI, deep multimodal learning, reinforcement learning, and attention mechanisms, need to be reviewed more comprehensively. Here are examples: Traffic speed prediction of urban road network based on high importance links using XGB and SHAP; Deep multimodal learning for traffic speed estimation combining dedicated short-range communication and vehicle detection system data; Travel time prediction using gated recurrent unit and spatio-temporal algorithm; An extended continuum model with consideration of the self-anticipative effect.

Experimental design

4. The classification of traffic states typically relies on only one or two indicators. While this study utilizes eight indicators, there is no justification provided for the selection of these particular indicators. Specifically, CO emissions and noise are results of congestion, not causes of it.
5. All the figures are difficult to interpret and understand. It is necessary to improve the readability by including more labels and text in the figures.
6. The descriptions accompanying the figures and tables are too brief. Clearly explain the results and discuss the implications of these findings in more detail.
7. This study uses a variety of models, but it does not explain how these models fit together into a cohesive framework.
8. In section 5.2, traffic state classification should be based on traffic flow properties that change over time and space. Simply clustering into k groups is meaningless from a traffic engineering perspective.
9. At a minimum, the study should include a fundamental diagram or similar method to present the traffic state data.
10. Although this study develops a model for predicting traffic states, it lacks any fundamentals from traffic flow theory.
11. Basic performance measures were not used. For class predictions, accuracy and F-1 score should be included, and for continuous value predictions, MAPE, MAE, and RMSE should be used to compare and evaluate model performance.

Validity of the findings

12. The policy implications of using the proposed model should also be discussed.
13. There are numerous typos throughout the manuscript. The authors should thoroughly proofread the document.
14. Overall, this study proposes a method for predicting traffic states using complex models and various measurable indicators. However, the structure of the model is not clearly explained, and there is insufficient justification for the selection of indicators. It may be better to focus on a single indicator, such as traffic speed, and use input variables like traffic data from time periods t-2, t-1, and t to predict traffic information for time period t+1.

Annotated reviews are not available for download in order to protect the identity of reviewers who chose to remain anonymous.

---

## Round 0.2 · Minor Revisions

Dear Authors,,

Thank you for the submission. The reviewers’ comments are now available. It is still not suggested that your article be published in its current format. We do, however, advise you to revise the paper in light of the reviewers’ comments and concerns before resubmitting it.

Best wishes,

·

Basic reporting

--

Experimental design

--

Validity of the findings

--

Additional comments

The authors have not addressed sufficient comments as suggested, especially:

5. The main drawback of this paper is related to an absence of sufficient details about the proposed methodology as the authors provided only some few details. The authors were invited to add theoretical part, some explaining flow charts, illustrations and algorithms.

However, the authors response is not sufficient.

Reviewer 3 ·

Basic reporting

I am happy with the responses.

Experimental design

The experimental design is well-constructed, and the revisions have made the model explanation clearer and easier to understand.

Validity of the findings

This study makes contributions by developing a model framework for predicting traffic conditions. The integration of classification models and the Transformer model is particularly innovative.

---

## Round 0.3 · Minor Revisions

Dear Authors,

Your "response to reviewer" and "tracked changes" files belong to the previous round. Please pay special attention for the criticisms and minor concerns provided for the last revision and resubmit the correct files.

Best wishes.

As a reminder, those comments were:

"The authors have not addressed sufficient comments as suggested, especially:
5. The main drawback of this paper is related to an absence of sufficient details about the proposed methodology as the authors provided only some few details. The authors were invited to add theoretical part, some explaining flow charts, illustrations and algorithms.
However, the authors response is not sufficient."

---

## Round 0.4 · accepted · Accept

Dear Authors,

Thank you for addressing the reviewers" comments. The paper seems sufficiently improved and ready for publication.

Best wishes,

·

Basic reporting

--

Experimental design

--

Validity of the findings

--

Additional comments

The authors have addressed all my comments as suggested.